# HMOs Induce Butyrate Production of *Faecalibacterium prausnitzii* via Cross-Feeding by *Bifidobacterium bifidum* with Different Mechanisms for HMO Types

**DOI:** 10.3390/microorganisms13071705

**Published:** 2025-07-21

**Authors:** Haruka Onodera, Yohei Sato, Yosuke Komatsu, Makoto Yamashita, Yuta Watanabe, Takeshi Kokubo

**Affiliations:** 1Institute of Health Sciences, Kirin Holdings Co., Ltd., 2-26-1-12-12 Muraoka-Higashi, Fujisawa 251-8555, Japan; haruka_onodera@kirin.co.jp (H.O.); yohei_sato@kirin.co.jp (Y.S.); yosuke_komatsu@kirin.co.jp (Y.K.); yuta_watanabe@kirin.co.jp (Y.W.); 2Kirin Central Research Institute, Kirin Holdings Co., Ltd., 2-26-1-12-12 Muraoka-Higashi, Fujisawa 251-8555, Japan; makoto_yamashita@kirin.co.jp

**Keywords:** human milk oligosaccharides, *Bifidobacterium bifidum*, *Faecalibacterium prasunitzii*, cross-feeding, butyrate

## Abstract

Human milk oligosaccharides (HMOs) have garnered significant attention as one of the bioactive components in human milk, with growing applications in infant formula and food products. HMOs enhance butyrate production, which is produced by butyrate-producing bacteria such as *Faecalibacterium prausnitzii* and contributes to gut health through its diverse biological functions. However, the specific mechanisms by which individual HMOs promote butyrate production remain unclear. In this study, we conducted in vitro co-culture experiments of *F. prausnitzii* and *Bifidobacterium bifidum*, examining their relative abundance, fatty acid production, residual sugar levels, and gene expression. Our results revealed that *B. bifidum* utilizes HMOs and provides the constituent sugars to *F. prausnitzii*, thereby promoting butyrate production by *F. prausnitzii*. Furthermore, we found that the underlying mechanisms vary depending on the structure of the HMOs. Specifically, 2′-fucosyllactose and 3′-sialyllactose enhance the butyrate production efficiency of *F. prausnitzii*, while 6′-sialyllactose primarily promotes the growth of *F. prausnitzii*. These findings not only deepen our understanding of how HMOs influence infant gut health but also suggest new directions for developing nutritional products that leverage the distinct functional properties of each HMO.

## 1. Introduction

Breastfeeding is considered the gold standard for infant nutrition, but infant formula is used when human milk is unavailable or cannot be provided [1]. Research and development efforts have aimed to make infant formula more similar to human milk; however, the addition of nutrients unique to human milk remains insufficient. Among the distinct components, human milk oligosaccharides (HMOs) have recently garnered attention [2]. HMOs are complex oligosaccharides found in human milk. They are composed of five types of monosaccharides: glucose (Glc), galactose (Gal), N-acetylglucosamine (GlcNAc), fucose (Fuc), and sialic acid (SIA). These monosaccharides combine in various lengths and sequences to form over 200 structurally diverse HMOs, each believed to have different biological functions [2,3]. In recent years, the roles of certain HMOs, such as 2′-fucosyllactose (2′-FL), 3′-sialyllactose (3′-SL), and 6′-sialyllactose (6′-SL), have been elucidated, supporting their incorporation in infant formula. Structurally, 2′-FL comprises fucose bound to lactose (Lac), while 3′-SL and 6′-SL contain SIA bound to Lac in distinct configurations (Figure 1). Moreover, these HMOs may offer health benefits not only for infants but also for children and adults [4,5,6], promoting their application in various food products [7].

One of the primary biological functions of HMOs is to promote the growth of *Bifidobacterium* species in the human colon. Infants who consume formula supplemented with HMOs develop a *Bifidobacterium*-dominant microbiota similar to that observed in breastfed infants [8,9]. In vitro studies have also confirmed that individual HMOs stimulate the growth of *Bifidobacterium* species [10,11]. Transporters and metabolic enzymes responsible for HMO utilization in *Bifidobacterium* species have been identified, which function in a specific HMO structure-dependent manner [12]. 2′-FL is degraded by fucosidase into Fuc and Lac, while 3′-SL and 6′-SL are degraded by sialidase into SIA and Lac. In addition to shaping the gut microbiota, HMOs promote the production of short-chain fatty acids (SCFAs) in the infant gut [13,14]. SCFAs are the key metabolites produced by the gut microbiome and play a crucial role in maintaining gut integrity and regulating various host physiological functions [15,16]. Among these, butyrate, produced by butyrate-producing bacteria, serves multiple important roles, including energy to colonic epithelial cells [17], modulating immune response via regulatory T cells [18], and strengthening the gut barrier [19]. Low intestinal butyrate levels in infants have been associated with a higher risk for developing allergies [20,21]. Moreover, HMO supplementation has been shown to elevate SCFA levels, including butyrate, in the fecal culture experiments of both children and adults [4,5,6], indicating their potential health benefits across a broad age range.

However, many butyrate-producing bacteria exhibit limited or no ability to metabolize HMOs [22,23,24]. Therefore, the growth and butyrate production in these bacteria may be promoted primarily via mechanisms other than direct HMO utilization. Previous studies have focused mainly on the effect of HMO mixtures on the overall gut microbiota [4,5,6,25]. Conversely, few studies have investigated the mechanisms underlying butyrate production, focusing on individual HMOs and specific butyrate-producing bacteria [22]. Gaining a better understanding of how individual HMOs promote butyrate production could shed light on the role of human milk in defining infant gut health. Ultimately, it may contribute to developing nutritional products that reflect the functional properties of specific HMOs.

The aim of this study was to elucidate the mechanisms by which individual HMOs enhance butyrate production. We focused on two bacterial strains commonly found in the human gut: *Bifidobacterium bifidum*, which possesses extracellular HMO-degrading enzymes [26,27], and *Faecalibacterium prausnitzii*, a highly abundant butyrate-producing bacterium [28]. *F. prausnitzii* also contributes to human health by generating butyrate using acetate produced by *Bifidobacterium* species [29,30]. In this study, we investigated the cross-feeding interaction between these two bacteria in the presence of 2′-FL, 3′-SL, and 6′-SL.

## 2. Materials and Methods

### 2.1. Chemicals and Bacterial Strains

Unless otherwise stated, all chemical reagents were purchased from Fujifilm Wako Pure Chemical Corporation (Osaka, Japan). HMOs (2′-FL, 3′-SL, and 6′-SL) and SIA were procured from Kyowa Hakko Bio Co., Ltd. (Tokyo, Japan).

*Bifidobacterium bifidum* JCM1254 and *Faecalibacterium prausnitzii* A2-165 (JCM31915) were provided by the Japan Collection of Microorganisms, RIKEN BRC, which participates in the National BioResource Project of the MEXT (Tsukuba, Japan).

### 2.2. Growth and Co-Culture Experiments

*B. bifidum* and *F. prausnitzii* were cultured in YCFA medium [31] supplemented with glucose as the sole energy source (YCFAG) and incubated anaerobically at 37 °C for 24 h using a Whitley A25 Workstation (Don Whitley Scientific, Bingley, UK). Next, the cultures were washed and resuspended in YCFA medium lacking sugars and VFA mix (YCFA-). Each bacterial suspension was adjusted to an OD_600_ of 1.0 with YCFA- medium. For the co-culture experiment, an equal volume of each suspension was inoculated into YCFA medium supplemented with 0.5% (*w*/*v*) of each carbohydrate at a ratio of 1:100. Cultures were incubated at 37 °C for 4–24 h anaerobically. The growth was measured as OD_600_ using Spectra Max (Molecular Devices, LLC., San Jose, CA, USA) for mono-cultures and CO7500 Colour Wave (Biochrom Ltd., Cambridge, UK) for co-culture experiments. Following incubation, the culture was harvested and centrifuged at 21,000× *g* for 5 min at 4 °C. The supernatants and the pellets were stored separately at −20 °C until further analysis.

### 2.3. Quantitative PCR (qPCR)

DNA was extracted from the bacterial pellets using ISOSPIN Fecal DNA (Nippon Gene Co., Ltd., Tokyo, Japan). The relative abundance of *B. bifidum* and *F. prausnitzii* was quantified by qPCR. The reaction mixture was prepared using GoTaq Green Master Mix (Promega K.K., Madison, WI, USA), and qPCR was performed with Quant Studio 3 (Thermo Fisher Scientific, Waltham, MA, USA). The previously reported primers were used: FPR-2F (5′-GGAGGAAGAAGGTCTTCGG-3′) and Fprau645R (5′-AATTCCGCCTACCTCTGCACT-3′) for *F. prausnitzii* and BiBIF-1 (5′-CCACATGATCGCATGTGATTG-3′) and BiBIF-2 (5′-CCGAAGGCTTGCTCCCAAA-3′) for *B. bifidum* [32]. Known concentrations of DNA samples extracted from each strain were used as standards. The qPCR cycling conditions were 95 °C for 20 s, followed by 40 cycles of 95 °C for 5 s, 55 °C for 30 s, and 72 °C for 45 s. The fluorescent signals were measured during the final extension step of each cycle.

### 2.4. Metabolite Analysis

The fatty acid composition in the culture supernatants was labeled using labeling reagent (XSRFAR, YMC Co., Ltd., Kyoto, Japan) and analyzed by HPLC, LC-20 Series (Shimadzu Corporation, Kyoto, Japan) equipped with a YMC-Pack FA column (YMC Co., Ltd.). 2-ethylbutyrate was used as an internal standard to normalize the data in each run. Two mobile phases were used: A (acetonitrile/methanol/H_2_O = 30/16/54) and B (acetonitrile/methanol/H_2_O = 20/16/64). The time program was as follows: A:B ratio = 0:100 (0–30 min); 100:0 (30–40 min); and 0:100 (40–45 min). The flow rate was 1 mL/min, with the column temperature set at 50 °C. The standard curve was encoded, and each peak was integrated using the LCsolution software (Shimadzu Corporation, Kyoto, Japan, ver. 3.50).

### 2.5. Sugar Concentration Analysis

The concentration of sugars in the culture supernatants was analyzed using a Dionex ICS-6000 HPIC system (Thermo Fisher Scientific) equipped with a Dionex CarboPac PA1 IC column (4 × 250 mm) (Thermo Fisher Scientific). Three mobile phases were used: A (H_2_O), B (300 mM CH_3_COONa), and C (500 mM NaOH). The programs were as follows: A:B:C ratio = 97.3:0.7:2 (0–30 min); 72:0:28 (30–35 min); 72:0:28 to 50:22:28 (35–43.5 min); 50:22:28 (43.5–50 min); 50:22:28 to 0:65:35 (50–55 min); 0:65:35 (55–60 min); and 97.3:0.7:2 (60–65 min). The flow rate was 1 mL/min, with the column temperature set at 25 °C. The standard curve was encoded, and each peak was integrated using the Chromeleon software (Thermo Fisher Scientific, Waltham, MA, USA, ver. 7.3).

### 2.6. RNA Sequencing (RNA-Seq)

*B. bifidum* and *F. prausnitzii* were cultured in YCFA medium supplemented with 0.5% (*w*/*v*) of different carbohydrates as described above until the midlog phase. The samples were immediately pelleted at 5000× *g* for 5 min at 4 °C and stored in RNAprotect Bacteria Reagent (Qiagen, Valencia, CA, USA). The samples were then centrifuged (5000× *g*, 4 °C, 10 min) and treated with tris-EDTA buffer (pH 8.0, Sigma-Aldrich Inc., St. Louis, MO, USA) containing 1 mg/mL lysozyme and >60 mAU/mL proteinase K (Qiagen) at room temperature for 10 min at 500 rpm. The total RNA was purified using the RNeasy Mini Kit (Qiagen) according to the manufacturer’s protocol.

The RNA-seq analysis was outsourced to Takara Bio Inc. (Kusatsu, Japan). The analysis included the following steps: the quality check of all reads was conducted using fastp (version 0.23.2), followed by the removal of low-quality reads. Filtering was performed using the default settings. After filtering, each read was mapped to the reference genome: GCF_024341745.1_ASM2434174v1_genomic.fna for *B. bifidum* and GCF_010509575.1_ASM1050957v1_genomic.fna for *F. prausnitzii*, using STAR (version 2.7.10a), with mapping conducted using STAR’s default settings. FeatureCounts (version 2.0.1) was used to calculate the count numbers for each gene. For the co-culture samples, counting was performed based on the RefSeq Gene ID as defined in the genomic.gtf. The reference genome and GTF files used were a combination of the two bacteria. The expression count data were normalized, and the CPM, FPKM, and TPM values were calculated.

### 2.7. Statistical Analysis

Unless otherwise stated, all data were obtained in triplicate for each experiment and expressed as mean ± standard deviation. Statistical significance was determined using one-way ANOVA with the Tukey–Kramer post hoc test and Dunnett’s test using the Bellcurve for Excel version 4.07 (**: *p* < 0.01).

## 3. Results

### 3.1. Investigation of Sugar Utilization by F. prausnitzii and B. bifidum

*F. prausnitzii* was unable to grow on any of the HMOs but grew on the constituent sugars of HMOs, except for Fuc (Appendix A). Furthermore, the growth of *F. prausnitzii* was significantly enhanced when acetate was combined with the HMOs constituent sugars, Glc, Gal, and Lac (Appendix A). *B. bifidum* grew in the presence of all tested HMOs. Among the constituent sugars, it grew on Glc or Lac, showed weak growth on Gal, and did not grow on Fuc or SIA (Appendix A).

### 3.2. Growth of F. Prausnitzii and B. bifidum Under the Co-Culture Condition

In a medium without any sugar source, neither *F. prausnitzii* nor *B. bifidum* exhibited growth in mono-culture or co-culture, and the ratio of the two bacteria remained nearly constant (Figure 2A,F). In the presence of Glc, growth was observed in the mono-cultures of each bacterium (Figure 2B). The co-culture condition led to a synergistic increase in OD_600_, and *B. bifidum* gradually became dominant (Figure 2F). In the presence of 2′-FL and 3′-SL, *B. bifidum* mono-culture and the co-culture showed increased OD_600_ (Figure 2C,D). Co-culturing again led to a synergistic growth with *B. bifidum* becoming dominant over time, similar to the Glc condition (Figure 2F). Notably, the relative abundance of *F. prausnitzii* remained higher with 3′-SL compared to Glc and 2′-FL. In contrast, in the presence of 6′-SL, co-culturing resulted in a lower OD_600_ than *B. bifidum* mono-culture after 12 h of culture, with *F. prausnitzii* becoming dominant (Figure 2E,F).

### 3.3. Fatty Acid Levels in the Co-Culture of F. prausnitzii and B. bifidum

We quantified the butyrate levels and simultaneously assessed lactate and acetate, which contribute to butyrate synthesis. Lactate and acetate were not detected in *F. prausnitzii* mono-culture under any sugar condition. Whereas, in *B. bifidum* mono-culture, these acids were detected under all sugar conditions tested. In the co-culture, lactate and acetate were detected under Glc, 2′-FL, and 3′-SL, though at lower levels than in the *B. bifidum* mono-culture (Figure 3A,B).

Butyrate was produced by *F. prausnitzii* mono-culture with Glc and in co-culture conditions with all sugar sources (Figure 3C). The amount of butyrate was similar for 2′-FL and 3′-SL but significantly higher in the presence of 6′-SL. However, upon normalization with the genome copy of *F. prausnitzii* (based on the qPCR results from Figure 2), butyrate production was notably lower with 6′-SL compared to 2′-FL and 3′-SL (Figure 3D).

### 3.4. Residual Sugar Levels in the Co-Culture of F. prausnitzii and B. bifidum

The concentrations of HMOs and their constituent sugars, Glc, Gal, Lac, Fuc, and SIA, were measured over time in the culture conditions. Glc was not detected in all conditions tested. 2′-FL remained unchanged in the *F. prausnitzii* mono-culture but was completely consumed within 24 h in both the *B. bifidum* mono-culture and co-culture, with Fuc released (Figure 4A,B). Lac and Gal were detected at lower concentrations and became undetectable at 24 h (Figure 4C,D). 3′-SL remained unchanged in the *F. prausnitzii* mono-culture but was fully consumed within 24 h in the *B. bifidum* mono-culture, releasing SIA (Figure 4E,F). In the co-culture, 3′-SL was also completely consumed by 24 h, though the amount of free SIA decreased by 1–2 mM compared to the degraded 3′-SL. Lac levels were similar between the *B. bifidum* mono-culture and co-culture, but Gal levels were lower in the co-culture than in the mono-culture (Figure 4G,H). 6′-SL showed a distinct pattern. It was not consumed by *F. prausnitzii* mono-culture but entirely in the *B. bifidum* mono-culture within 24 h, though more slowly than 2′-FL and 3′-SL. In the co-culture, nearly 80% of 6′-SL remained after 24 h (Figure 4I). SIA, Lac, and Gal were detected in the *B. bifidum* mono-culture but were absent at any time point during co-culture (Figure 4J–L).

### 3.5. Gene Expression Analysis in the Co-Culture of F. prausnitzii and B. bifidum

In both *F. prausnitzii* and *B. bifidum*, gene expression patterns under HMO treatment showed the strongest correlation between 2′-FL and 3′-SL, followed by a high correlation between 3′-SL and 6′-SL (Figure 5A–F). HMOs consistently induced β-galactosidases expression, a key enzyme of Lac metabolism (Figure 5G). *F. prausnitzii* showed the upregulation of genes involved in Gal metabolism under all HMOs, while *B. bifidum* exhibited a weaker response. Genes involved in Glc metabolism were generally upregulated in *F. prausnitzii*, particularly under 6′-SL. In contrast, *B. bifidum* showed variable expression with no significant differences across the HMOs. The SIA metabolism enzymes were downregulated in *F. prausnitzii* under 6′-SL.

## 4. Discussion

In recent years, growing evidence has supported the role of HMOs in promoting the production of SCFAs, including butyrate. However, only a few butyrate-producing bacteria can directly metabolize HMOs. The mechanisms by which individual HMOs contribute to butyrate production remain largely unclear. In this study, we investigated how 2′-FL, 3′-SL, and 6′-SL promote butyrate production via cross-feeding during the co-culture of *B. bifidum* and *F. prausnitzii*.

To understand this interaction, we first evaluated the ability of *F. prausnitzii* A2-165 and *B. bifidum* JCM1254 to utilize sugars related to HMOs. *F. prausnitzii* did not metabolize HMOs themselves but was able to utilize the constituent sugars such as Lac, Gal, Glc, and SIA. We also confirmed that the growth of *F. prausnitzii* was significantly promoted in the presence of acetate, a precursor of butyrate, along with these sugars. Conversely, *B. bifidum* could utilize HMOs as well as Lac, Gal, and Glc but not Fuc or SIA. These patterns are consistent with the results of previous studies [22,31,33,34,35].

Next, we examined co-cultures of *F. prausnitzii* and *B. bifidum* in the presence of HMOs. This led to increased butyrate production by *F. prausnitzii*. Given its inability to directly utilize HMOs, this phenomenon suggests that *F. prausnitzii* utilized the HMOs’ constituent sugars that were released by the extracellular enzymatic activity of *B. bifidum* [26,27]. The results indicate the occurrence of cross-feeding of HMOs from *B. bifidum* to *F. prausnitzii*. Previous studies have shown that co-culturing *F. prausnitzii* and *B. infantis* does not promote butyrate production in the presence of HMOs [22]. Since *B. infantis* degrades HMOs intracellularly, these results suggest the importance of extracellular degradation of HMOs for effective cross-feeding.

When co-cultured in the presence of 2′-FL, *B. bifidum* eventually became dominant, corresponding with its increased production of lactate and acetate. Despite the low abundance of *F. prausnitzii*, a significant amount of butyrate was produced. Additionally, the amount of acetate, a precursor of butyrate, was lower in co-culture than in *B. bifidum* mono-culture. These observations suggest that 2′-FL is degraded to Fuc, Lac, Gal, and Glc by the extracellular enzymes of *B. bifidum* [26]. While most of these degraded sugars are consumed by *B. bifidum*, *F. prausnitzii* utilizes some of them along with acetate, achieving high butyrate production even at low population levels [31,36]. *B. bifidum* in mono-culture has a low ability to utilize Gal; however, the Gal metabolism was highly upregulated in *F. prausnitzii* when co-cultured with *B. bifidum*. This suggests that *F. prausnitzii* utilized Gal with a competitive advantage.

Similar patterns were observed with 3′-SL, where co-culturing resulted in dominance of *B. bifidum*, increased butyrate production, decreased acetate, and elevated Gal metabolism gene expression in *F. prausnitzii*. Although the degradation rate of 3′-SL was similar in the mono-culture and co-culture of *B. bifidum*, the amount of free SIA was reduced by 1–2 mM in co-culture. Since *B. bifidum* cannot metabolize SIA, this reduction is likely due to utilization by *F. prausnitzii*. Therefore, despite the dominance of *B. bifidum*, *F. prausnitzii* efficiently produced butyrate by utilizing 3′-SL component sugars such as Gal and SIA as well as acetate produced by *B. bifidum* [31,36]. Although the overall trends under 2′-FL and 3′-SL were similar, compared to 2′-FL, co-culture under 3′-SL showed a higher proportion of *F. prausnitzii* and lower production of lactate and acetate. These differences are possibly due to the presence of SIA.

6′-SL showed distinct characteristics compared with 2′-FL and 3′-SL. In co-culture, the total OD_600_ was markedly reduced compared to *B. bifidum* mono-culture. *F. prausnitzii* became dominant, eventually comprising over 90% of the population. Consistently, lactate and acetate, typically produced by *B. bifidum*, were not detected, and only about 20% of 6′-SL was consumed after 24 h. The butyrate production efficiency of *F. prausnitzii* was likely low due to the limited supply of acetate, a precursor of butyrate, from *B. bifidum*, leading to reduced activity of the butyrate production pathway. Although a certain amount of 6′-SL was degraded, the constituent sugars were not detected at any time point. The metabolic pathways of Glc, Gal, and Lac were upregulated in *F. prausnitzii*. This indicates that these sugars were released and immediately consumed by *F. prausnitzii*. Conversely, the expression of SIA metabolism enzymes was suppressed compared with the Glc-supplemented condition, despite the lack of detectable SIA. Since the expression of SIA metabolism enzymes can be transient [37], the timing of the evaluation might not have coincided with the timing of SIA consumption. Time-course analysis with more time points could provide further insights.

Although 3′-SL and 6′-SL are structurally similar, they showed completely different trends. Gene expression analysis revealed that 3′-SL induced a response more similar to 2′-FL than to 6′-SL. This difference is likely due to the varying rates of HMO degradation by *B. bifidum*. Although the structural basis is not well understood, a previous study reported that the sialidase of *B. bifidum* degrades 6′-SL less efficiently compared to 3′-SL [27], consistent with our sugar analysis results. Rapid degradation of 2′-FL and 3′-SL led to the early activation of *B. bifidum* metabolism, allowing it to become dominant over *F. prausnitzii*. In contrast, slower degradation of 6′-SL allowed *F. prausnitzii* to utilize the released sugars first, thereby becoming more dominant. This is supported by the enhanced sugar metabolism observed in *F. prausnitzii* under 6′-SL-supplemented conditions. These results may imply that 2′-FL and 3′-SL are potentially useful for increasing butyrate production efficiency of *F. prausnitzii* in individuals with a pre-existing and sufficient abundance of this bacterium. Meanwhile, 6′-SL might promote the growth of *F. prausnitzii* in individuals with lower levels.

One limitation of this study is the challenge in evaluating the specific consumption of Glc, Gal, and Lac by each bacterium in the co-culture system. Additionally, this study focused on *B. bifidum* as an HMO-degrader and *F. prausnitzii* as a butyrate-producer; thus, the insights could be limited to these particular strains. In the human gut, other bacteria such as *Bacteroides* and *Akkermansia* also act as extracellular HMO degraders [38,39]. Their HMO-degrading enzymes differ in their substrate specificity and efficiency [40]. Similarly, other butyrate-producers, such as *Eubacterium* and *Roseburia*, differ in their ability to metabolize HMOs and their constituent sugars [23,41]. Therefore, cross-feeding can be strain-dependent, and diverse bacterial combinations may contribute to butyrate production in the human body. Evaluating different combinations of bacterial strains could help elucidate the broader mechanism modulating this process. The gut microenvironment involves complex host–microbe interactions. Hence, it is also necessary to evaluate whether the metabolic features observed between the two strains in this study can be reproduced in models for investigating host–microbe interactions, such as gnotobiotic mouse models.

## 5. Conclusions

This study demonstrated that *B. bifidum* promotes butyrate production by *F. prausnitzii* through the cross-feeding of HMOs and that the mechanisms vary depending on the type of HMO. While 2′-FL and 3′-SL enhance the butyrate production efficiency, 6′-SL increases the number of butyrate-producing bacteria. These findings improve our understanding of how individual HMOs impact the infant gut health and will guide in developing nutritional products that utilize the unique characteristics of each HMO. Furthermore, the symbiotic administration of *B. bifidum* and HMOs may enhance butyrate production by *F. prausnitzii* in the gut via HMO cross-feeding, potentially supporting butyrate-driven health promotion.

## Figures and Tables

**Figure 1 microorganisms-13-01705-f001:**
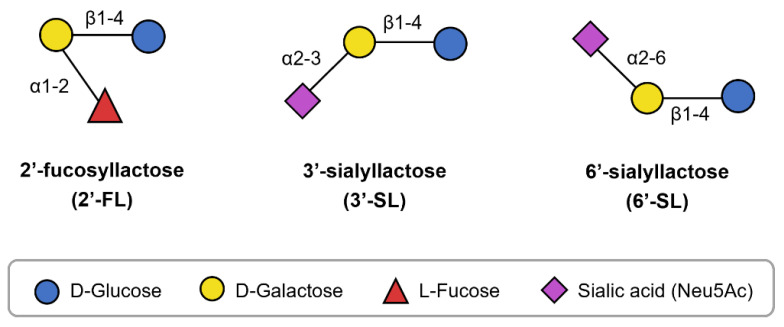
Structure of 2′-FL, 3′-SL, and 6′-SL.

**Figure 2 microorganisms-13-01705-f002:**
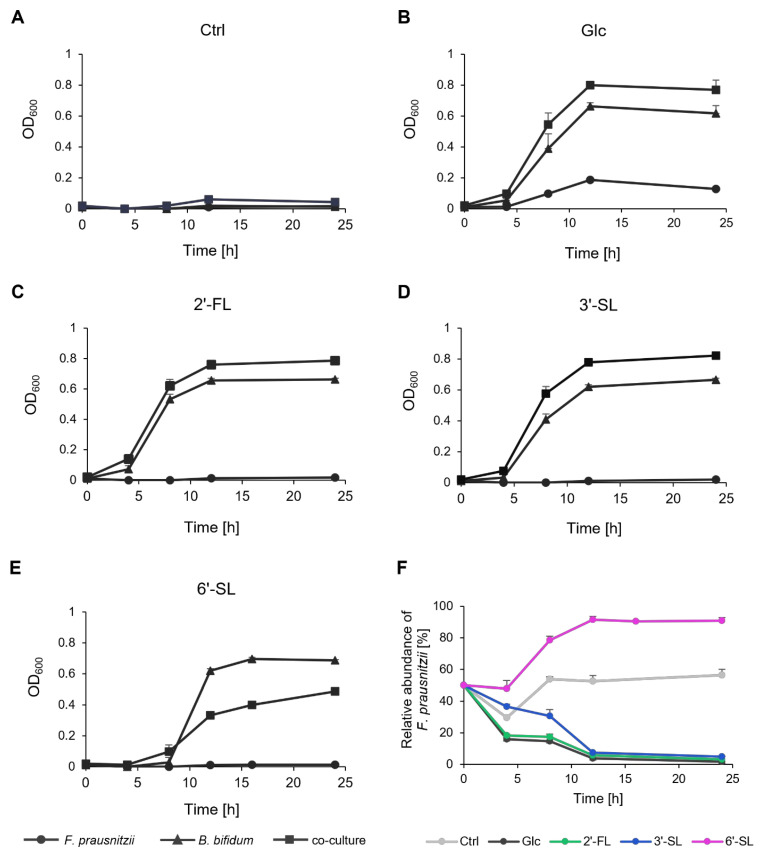
Growth and relative abundance of *F. prausnitzii* and *B. bifidum* in mono-culture and co-culture with individual sugars. (**A**–**E**) The growth curve of *F. prausnitzii* and *B. bifidum* in mono-culture and co-culture under (**A**) no-sugar (Ctrl), (**B**) Glc, (**C**) 2′-FL, (**D**) 3′-SL, and (**E**) 6′-SL. Symbols represent mono-culture of *F. prausnitzii* (circle), mono-culture of *B. bifidum* (triangle), and co-culture (rectangle). (**F**) Relative abundance of *F. prausnitzii* in co-culture with *B. bifidum*. The values at 0 h are the calculated theoretical values. Data are shown as the means ± standard deviation (*n* = 3). Ctrl, control; Glc, glucose; 2′-FL, 2′-fucosyllactose; 3′-SL, 3′-sialyllactose; 6′-SL, 6′-sialyllactose.

**Figure 3 microorganisms-13-01705-f003:**
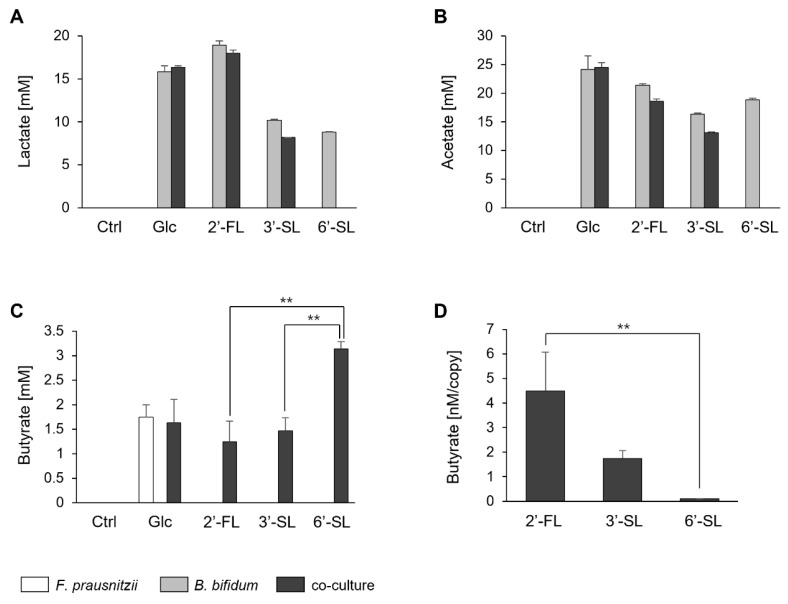
Fatty acid production by *F. prausnitzii* and *B. bifidum* in mono-culture and co-culture with individual sugars. (**A**–**C**) Fatty acid profiles after 24 h of mono-culture and co-culture under no-sugar (Ctrl), Glc, 2′-FL, 3′-SL, and 6′-SL. Bars represent the profiles of mono-culture of *F. prausnitzii* (white), mono-culture of *B. bifidum* (gray), and co-culture (black). (**D**) Butyrate production efficiency of *F. prausnitzii* under 2′-FL, 3′-SL, and 6′-SL conditions, calculated by dividing the amount of butyrate by the genome copy number of *F. prausnitzii* (quantified by qPCR). Data are shown as the means ± standard deviation (*n* = 3). Statistical comparisons among the sugar conditions were performed using Tukey’s test. Ctrl, control; Glc, glucose; 2′-FL, 2′-fucosyllactose; 3′-SL, 3′-sialyllactose; 6′-SL, 6′-sialyllactose. **: *p* < 0.01.

**Figure 4 microorganisms-13-01705-f004:**
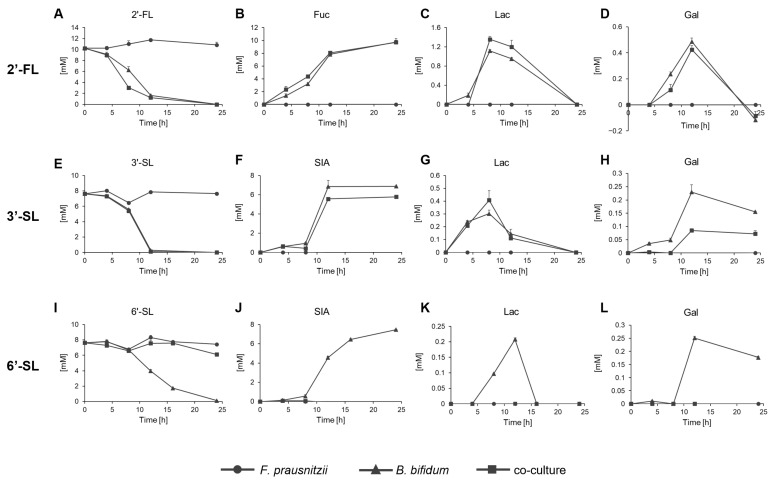
Residual sugar analysis of *F. prausnitzii* and *B. bifidum* in mono-culture and co-culture with individual sugars. Residual sugar profiles of *F. prausnitzii* and *B. bifidum* analyzed under (**A**–**D**) 2′-FL, (**E**–**H**) 3′-SL, and (**I**–**L**) 6′-SL. Symbols represent mono-culture of *F. prausnitzii* (circle), mono-culture of *B. bifidum* (triangle), and co-culture (rectangle). Data are shown as the means ± standard deviation (*n* = 3). 2′-FL, 2′-fucosyllactose; 3′-SL, 3′-sialyllactose; 6′-SL, 6′-sialyllactose; Fuc, fucose; SIA; sialic acid; Gal, galactose; Lac, lactose.

**Figure 5 microorganisms-13-01705-f005:**
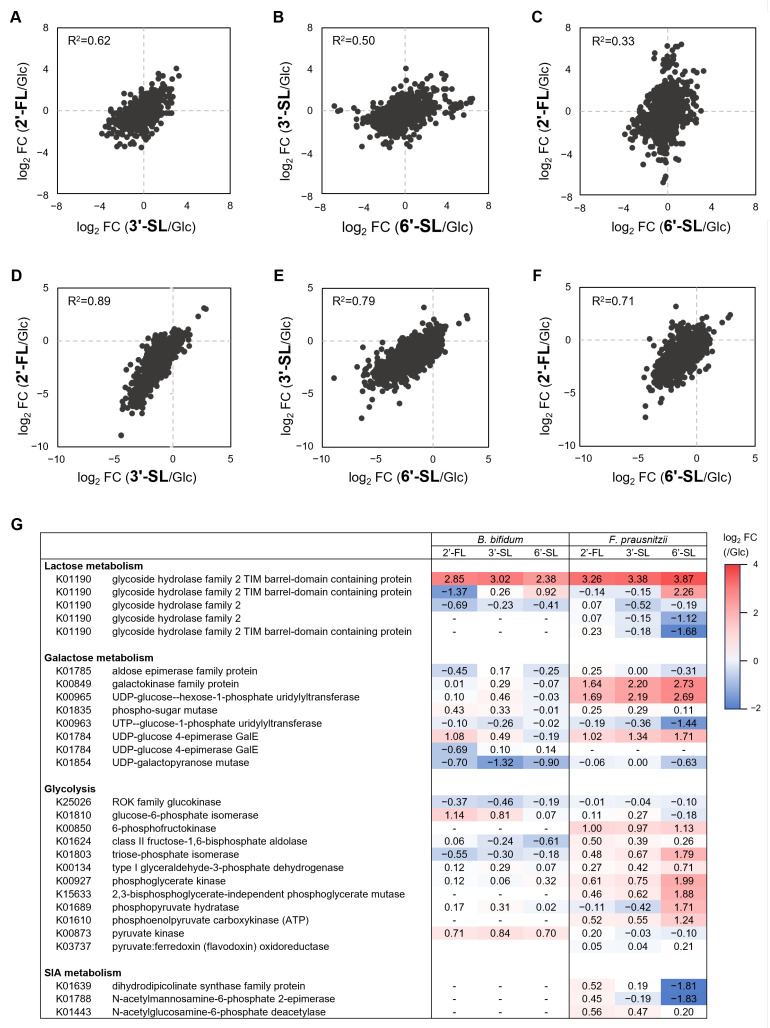
Gene expression analysis of *F. prausnitzii* and *B. bifidum* in co-culture with individual sugars. Two-dimensional plots of the fold change in gene expression under 2′-FL, 3′-SL, and 6′-SL compared to the Glc-supplemented condition in (**A**–**C**) *F. prausnitzii* and (**D**–**F**) *B. bifidum*. Each plot is represented with Pearson’s rho. (**G**) Heatmap representing the fold change values of gene expression of enzymes related to each sugar metabolism. RNA-seq data from a single replicate were used for the analysis. Glc, glucose; 2′-FL, 2′-fucosyllactose; 3′-SL, 3′-sialyllactose; 6′-SL, 6′-sialyllactose.

## Data Availability

The original contributions presented in this study are included in the article/Appendix A. Further inquiries can be directed to the corresponding author.

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
