# Peer review of "HMOs Induce Butyrate Production of Faecalibacterium prausnitzii via Cross-Feeding by Bifidobacterium bifidum with Different Mechanisms for HMO Types"

_microorganisms, 2025, doi:10.3390/microorganisms13071705_

Round 1
Reviewer 1 Report
Comments and Suggestions for Authors
The article presents a well-organized study of the role of human milk oligosaccharides (HMOs) in promoting butyrate production by cross-feeding interactions between Bifidobacterium bifidum and Faecalibacterium prausnitzii. This study is relevant to infant gut health and is of potential value for nutritional product development. The experimental approach includes growth assays, metabolite profiling, and gene expression analysis. Some methodological and discussion limitations should, however, be pointed out.
Comment 1
One major limitation, also mentioned by the authors, is that the study used only two strains, B. bifidum JCM1254 and F. prausnitzii A2-165, which limits generalizability. Have other HMO-degrading (e.g. B. infantis, which degrades HMOs intracellularly) or butyrate-producing (e.g. Roseburia) strains been tested, or is this planned for future work? It remains unclear whether the observed dynamics are limited to F. prausnitzii or represent a more general functional trait. Other strains may exhibit completely different cross-feeding behaviors.
Future research should investigate:
Whether B. infantis (or other HMO consumers) can similarly stimulate F. prausnitzii growth?
How other butyrate producers cooperate or interact within this niche?
Whether such pairwise interactions extend to complex microbial communities?
Comment 2
The findings are based on in vitro co-cultures, which do not fully reflect the complexity of the gut environment. While such experiments provide informative preliminary results, the absence of in vivo validation limits their potential physiological relevance. The gut microenvironment involves complex host-microbe interactions, spatial organization, and competitive pressures that are difficult to reproduce in simplified in vitro systems.
In my opinion, to strengthen the conclusions, future work should include the use of gnotobiotic mouse models to assess strain persistence, in vivo metabolic activity, and host responses.
Equally valuable would be experiments using human infant fecal cultures to quantify the stability of the observed interactions and their functional impact at the community level.
The authors should address this limitation in the discussion section and propose these validation methods as essential next steps in future research.
Comment 3
The authors should consider discussing whether their results support the selective enrichment of specific HMO structures to modulate microbial communities or metabolic outcomes. This would make the study more applicable to product development and nutrition research.
Comment 4
Could the authors clarify how extracellular enzymes differentially process 2′-FL, 3′-SL, and 6′-SL? Are there structural or kinetic determinants responsible for substrate preference?
Comment 5
Butyrate is normalized to F. prausnitzii genome copies (Fig. 3D), but cellular activity may vary depending on conditions. Could the lower efficiency with 6′-SL be due to slower growth rather than reduced metabolic output?
Additionally, why does 6′-SL-induced F. prausnitzii dominance not correlate with higher butyrate production per cell (Fig. 3D)?
Comment 6
Notably, 6'-SL promoted F. prausnitzii predominance and significantly inhibited B. bifidum growth, as demonstrated by both relative abundance and ~ 50% reduction in OD600 in co-culture compared to B. bifidum monoculture at 24 h. Such a competitive effect is an interesting finding that should be emphasized.
Comment 7
Lines 186-187
The statement about reduced OD600 in co-culture with 6′-SL should specify the relevant time point, as Figure 2E indicates comparable OD600 values in co-culture and B. bifidum mono-culture at earlier time points.
With revisions and clarification of key points, the manuscript will be suitable for publication.
Comments on the Quality of English LanguageThere are some minor grammatical errors, occasional awkward phrasing, and long sentences that disrupt readability. For example:
- B. bifidum cross-feeds on HMOs, thereby promoting butyrate production by F. prausnitzii
In my opinion should be
- B. bifidum utilizes HMOs through cross-feeding, enhancing butyrate production by F. prausnitzii.
Please proofread the text for clarity and grammar.
Author Response
Dear Reviewer,
Thank you very much for your valuable and insightful comments on our manuscript. We sincerely appreciate the time and effort you have devoted to reviewing our work. We have addressed your comments with point-by-point responses and have revised the manuscript accordingly.
Comment 1: One major limitation, also mentioned by the authors, is that the study used only two strains, B. bifidum JCM1254 and F. prausnitzii A2-165, which limits generalizability. Have other HMO-degrading (e.g. B. infantis, which degrades HMOs intracellularly) or butyrate-producing (e.g. Roseburia) strains been tested, or is this planned for future work? It remains unclear whether the observed dynamics are limited to F. prausnitzii or represent a more general functional trait. Other strains may exhibit completely different cross-feeding behaviors.
Future research should investigate:
Whether B. infantis (or other HMO consumers) can similarly stimulate F. prausnitzii growth?
How other butyrate producers cooperate or interact within this niche?
Whether such pairwise interactions extend to complex microbial communities?
Response 1: Thank you for your insightful comments. We agree that other bacteria may show different trends and that further studies are needed as mentioned in the Discussion.
It has been reported that co-culturing B. infantis and F. prausnitzii in the presence of HMOs does not enhance butyrate production as noted in line 292. Since HMO-degrading enzymes of B. infantis are intracellular, the degradation products are difficult for other microorganisms to utilize, and cross-feeding is likely inefficient. Therefore, we focus on Bacteroides and Akkermansia, which possess extracellular HMO-degrading enzymes like B. bifidum. Although preliminary, we have confirmed that these extracellular HMO degraders also promote butyrate production by F. prausnitzii through cross-feeding. We intend to further investigate the cross-feeding mechanisms and their differences from B. bifidum in future studies.
Comment 2: The findings are based on in vitro co-cultures, which do not fully reflect the complexity of the gut environment. While such experiments provide informative preliminary results, the absence of in vivo validation limits their potential physiological relevance. The gut microenvironment involves complex host-microbe interactions, spatial organization, and competitive pressures that are difficult to reproduce in simplified in vitro systems.
In my opinion, to strengthen the conclusions, future work should include the use of gnotobiotic mouse models to assess strain persistence, in vivo metabolic activity, and host responses.
Equally valuable would be experiments using human infant fecal cultures to quantify the stability of the observed interactions and their functional impact at the community level.
The authors should address this limitation in the discussion section and propose these validation methods as essential next steps in future research.
Response 2: Thank you for your suggestion. We agree that further evaluation is necessary to assess whether the metabolic features of the two strains can be reproduced using in vivo models involving host-microbe interactions. However, as mentioned in the Introduction section, some fecal culture studies have shown that HMOs increase butyrate-producing bacteria and butyrate production. The interaction between two specific strains could not be evaluated there because it is difficult to examine the metabolism of individual bacteria in the complex fecal culture system. The significance of our study lies in isolating and analyzing this interaction in detail. Therefore, we have added a discussion focusing on host-microbe interactions in the Discussion section.
Line 357: The gut microenvironment involves complex host-microbe interactions. Hence, it is also necessary to evaluate whether the metabolic features observed between the two strains in this study can be reproduced in models for investigating host-microbe interactions, such as gnotobiotic mouse models.
Comment 3: The authors should consider discussing whether their results support the selective enrichment of specific HMO structures to modulate microbial communities or metabolic outcomes. This would make the study more applicable to product development and nutrition research.
Response 3: I apologize if I have misunderstood the intent of your question, but I understood that you were asking for additional discussion on how individual HMOs may contribute to different states of microbial communities. Accordingly, I have added the following points to the Discussion section.
Line 342: These results may imply that 2′-FL and 3′-SL are potentially useful for increasing butyrate production efficiency of F. prausnitzii in individuals with a pre-existing and sufficient abundance of this bacterium. Meanwhile, 6′-SL might promote the growth of F. prausnitzii in individuals with lower levels.
Comment 4: Could the authors clarify how extracellular enzymes differentially process 2′-FL, 3′-SL, and 6′-SL? Are there structural or kinetic determinants responsible for substrate preference?
Response 4: 2′-FL is degraded by an enzyme different from the enzyme that degrades 3′-SL or 6′-SL. 2′-FL is degraded by fucosidase, while 3′-SL and 6′-SL are degraded by sialidase. Since this was not mentioned in the main text and the distinction was unclear, we have added this information.
Line 53: 2′-FL is degraded by fucosidase into Fuc and Lac, while 3′-SL and 6′-SL are degraded by sialidase into SIA and Lac.
As described in the Discussion section, another research group showed that sialidase has higher substrate specificity for 3’-SL than for 6’-SL. However, the structural basis remains unclear, making structural or kinetic explanations difficult. This information has also been added to our manuscript.
Line 336: Although the structural basis is not well understood, a previous study reported that the sialidase of B. bifidum degrades 6′-SL less efficiently compared to 3′-SL [27], consistent with our sugar analysis results.
Comment 5: Butyrate is normalized to F. prausnitzii genome copies (Fig. 3D), but cellular activity may vary depending on conditions. Could the lower efficiency with 6′-SL be due to slower growth rather than reduced metabolic output?
Additionally, why does 6′-SL-induced F. prausnitzii dominance not correlate with higher butyrate production per cell (Fig. 3D)?
Response 5: When co-cultured under the 6′-SL condition, F. prausnitzii became dominant and its carbohydrate metabolism was activated, indicating that its growth was promoted. Nevertheless, the efficiency of butyrate production was low. This is likely due to an insufficient supply of acetate, a precursor of butyrate, from B. bifidum, leading to reduced activity of the butyrate production pathway in F. prausnitzii. We have added this explanation to the text.
Line 322: The butyrate production efficiency of F. prausnitzii was likely low due to the limited supply of acetate, a precursor of butyrate, from B. bifidum, leading to reduced activity of the butyrate production pathway.
Comment 6: Notably, 6'-SL promoted F. prausnitzii predominance and significantly inhibited B. bifidum growth, as demonstrated by both relative abundance and ~ 50% reduction in OD600 in co-culture compared to B. bifidum monoculture at 24 h. Such a competitive effect is an interesting finding that should be emphasized.
Response 6: Although the distinctive trend of 6'-SL has already been discussed, we have explicitly stated this in the Discussion section, as you pointed out.
Line 319: In co-culture, the total OD600 was markedly reduced compared to B. bifidum mono-culture.
Comment 7: Lines 186-187
The statement about reduced OD600 in co-culture with 6′-SL should specify the relevant time point, as Figure 2E indicates comparable OD600 values in co-culture and B. bifidum mono-culture at earlier time points.
Response 7: As you pointed out, we have clearly added the relevant culture duration.
Line 190: In contrast, in the presence of 6′-SL, co-culturing resulted in a lower OD600 than B. bifidum mono-culture after 12 h of culture, with F. prausnitzii becoming dominant (Figure 2E, F).
Comments on the Quality of English Language: There are some minor grammatical errors, occasional awkward phrasing, and long sentences that disrupt readability. For example:
B. bifidum cross-feeds on HMOs, thereby promoting butyrate production by F. prausnitzii
In my opinion should be
B. bifidum utilizes HMOs through cross-feeding, enhancing butyrate production by F. prausnitzii.
Please proofread the text for clarity and grammar.
Response: To make the expression more direct, I revised the sentence as follows:
"B. bifidum utilizes HMOs and provides the constituent sugars to F. prausnitzii, thereby promoting butyrate production by F. prausnitzii."
The overall grammar has been checked by a native proofreader. Additionally, we have broken up long sentences and made other adjustments to improve clarity. We also made some minor corrections without changing the content, such as fixing typographical errors.
Once again, we are grateful for your constructive feedback, which has helped improve the clarity of our manuscript. We look forward to further advancing this work based on these suggestions.
Thank you for your kind consideration.
Sincerely,
Haruka Onodera
Reviewer 2 Report
Comments and Suggestions for Authors
The manuscript is interesting, technically well-developed and concise, which is appreciated given the enormous volume of scientific information available today. I find no impediment to recommending its publication, with a few minor changes related to formatting issues:
Lines 142-144: “g” when is related to centrifugation, should be written in italics to differentiate from grams abbreviation.
Line 169: “Faecalibacterium prausnitzii” was previously abbreviated as “F. prausnitzii”.
Lines 168-174: Each sugar, the first time that is cited in the text should be written by its complete name.
Lines 354-356: The relation with Akkermansia and Bacteroides are limited to a brief comment at the end of the discussion. I think that the degree of evidence related to the action of these two bacterial genera are not the conclusion of the current work. Thus, it would be deleted from the conclusions section, that must refer only to achievements of the current manuscript.
Author Response
Dear Reviewer,
Thank you very much for your valuable comments on our manuscript. We sincerely appreciate the time and effort you have devoted to reviewing our work. We have addressed your comments with point-by-point responses and have revised the manuscript accordingly.
Comment 1: Lines 142-144: “g” when is related to centrifugation, should be written in italics to differentiate from grams abbreviation.
Response 1: We have made the corrections as you pointed out (lines 147-148).
Comment 2: Line 169: “Faecalibacterium prausnitzii” was previously abbreviated as “F. prausnitzii”.
Response 2: We have made the corrections as you pointed out (line 173).
Comment 3: Lines 168-174: Each sugar, the first time that is cited in the text should be written by its complete name.
Response 3: We have added the explanation of lactose (Lac) in line 43: Structurally, 2′-FL comprises fucose bound to lactose (Lac), while 3′-SL and 6′-SL contain SIA bound to Lac in distinct configurations (Figure 1).
Other sugars are described in lines 37–38: HMOs are complex oligosaccharides found in human milk, composed of five types of monosaccharides: glucose (Glc), galactose (Gal), N-acetylglucosamine (GlcNAc), fucose (Fuc), and sialic acid (SIA).
Comment 4: Lines 354-356: The relation with Akkermansia and Bacteroides are limited to a brief comment at the end of the discussion. I think that the degree of evidence related to the action of these two bacterial genera are not the conclusion of the current work. Thus, it would be deleted from the conclusions section, that must refer only to achievements of the current manuscript.
Response 4: As we agree with your point, we have removed the mention of Akkermansia and Bacteroides (lines 370-371). Additionally, the conclusion contained overly detailed content, so we have adjusted it to be more appropriate.
In addition, we made some minor corrections without changing the content, such as fixing typographical errors.
Once again, we are grateful for your feedback, which has helped improve the clarity of our manuscript. We look forward to further advancing this work based on these suggestions.
Thank you for your kind consideration.
Sincerely,
Haruka Onodera